# CFD Simulation of a Submersible Passive Rotor at a Pipe Outlet under Time-Varying Water Jet Flux

**Mohamed Farouk** [1,2,*] , **Karim Kriaa** [1,3] and **Mohamed Elgamal** [1,4]

1   College of Engineering, Imam Mohammad Ibn Saud Islamic University, IMSIU, Riyadh 11432, Saudi Arabia
2   Irrigation and Hydraulics Department, Faculty of Engineering, Ain Shams University, Cairo 11517, Egypt
3   Department of Chemical Engineering, National School of Engineers of Gabes, University of Gabes, Gabes 6029, Tunisia
4   Irrigation and Hydraulics Department, Faculty of Engineering, Cairo University, Giza 12613, Egypt
*   Correspondence: miradi@imamu.edu.sa

**Abstract:** During the past two decades, passive rotors have been proposed and introduced to be used in a number of different water sector applications. One of these applications is the use of a passive rotor at the outlets of pipe outfalls to enhance mixing. The main objective of this study is to develop a CFD computational workflow to numerically examine the feasibility of using a passive rotor downstream of the outlet of pipe outfalls to improve the mixing properties of the near flow field. The numerical simulation for a pipe outlet with a passive rotor is a numerical challenge because of the nonlinear water-structure interactions between the water flow and the rotor. This study utilizes a computational workflow based on the ANSYS FLUENT to simulate that water-structure interaction to estimate the variation in time of the angular speed ($\omega$) of a passive rotor initially at rest and then subjected to time-varying water velocity ($\upsilon$). Two computational techniques were investigated: the six-degrees-of-freedom (6DOF) and the sliding mesh (SM). The 6DOF method was applied first to obtain a mathematical relation of $\omega$ as a function of the water velocity ($\upsilon$). The SM technique was used next (based on the deduced $\omega$-$\upsilon$ relation by the 6DOF) to minimize the calculation time considerably. The study has shown that the 6DOF technique accurately determines both maximum and temporal angular speeds, with discrepancies within 3% of the measured values. A number of numerical runs were conducted to investigate the effect of the gap distance between the passive rotor and the pipe outlet and to examine the effect of using the passive rotor on the near flow field downstream of the rotor. The model results showed that as the gap distance of the pipe outlet to the passive rotor increases, the rotor's maximum angular speed decreases following a decline power-law trend. The numerical model results also revealed that the passive rotor creates a spiral motion that extends downstream to about 15 times the pipe outlet diameter. The passive rotor significantly increases the turbulence intensity by more than 500% in the near field zone of the pipe outlet; however, this effect rapidly vanishes after four times the pipe diameter.

**Keywords:** turbulence closure; k-$\varepsilon$ model; varying bed topography; flow over bedforms; turbulence intensity

## 1. Introduction

Nowadays, over 150 countries utilize desalination in some form or another to meet their individual water needs and provide water to more than 300 million people. However, the effluent from those desalination facilities generally negatively impacts the environment and requires thermal or brine management [1,2].

The concept of utilizing passive rotors is found in the aeronautical engineering literature, where passive rotors are added to the original wide chord blade rotors to enhance the system's performance at large [3].

In the water engineering technology literature, a "passive rotor" generally refers to a rotor that naturally revolves due to the induced forces from the water–rotor interactions

(without any external power sources to operate) [4]. Passive rotors differ from water turbines in that they contain only the energy capture mechanism (the rotor blades system), whereas water turbines have both the energy capture and conversion mechanisms. Therefore, passive rotors are not subjected to external loads (from generators). Consequently, it will rotate at a velocity (called the "runaway velocity") primarily determined by the rotating torque created by the fluid on the blades and by the resistance generated by the liquid on the revolving blades.

During the last decade, passive rotors have been proposed/used in different water and environmental engineering applications. Figure 1 explores the samples of these applications, which include:

- Use of a passive rotor at the water tank pipe outlet to increase the effluent drainage rate from the effluent tanks in water treatment facilities [5];
- Use of a passive rotor at the outlet of pipe outfall to improve mixing;
- Using a passive rotor upstream of a water level control gate to adjust the upstream water afflux by controlling the rotation of the passive rotor without the need to change the gate opening [3].
- Using passive rotors downstream of a sluice gate for energy dissipation

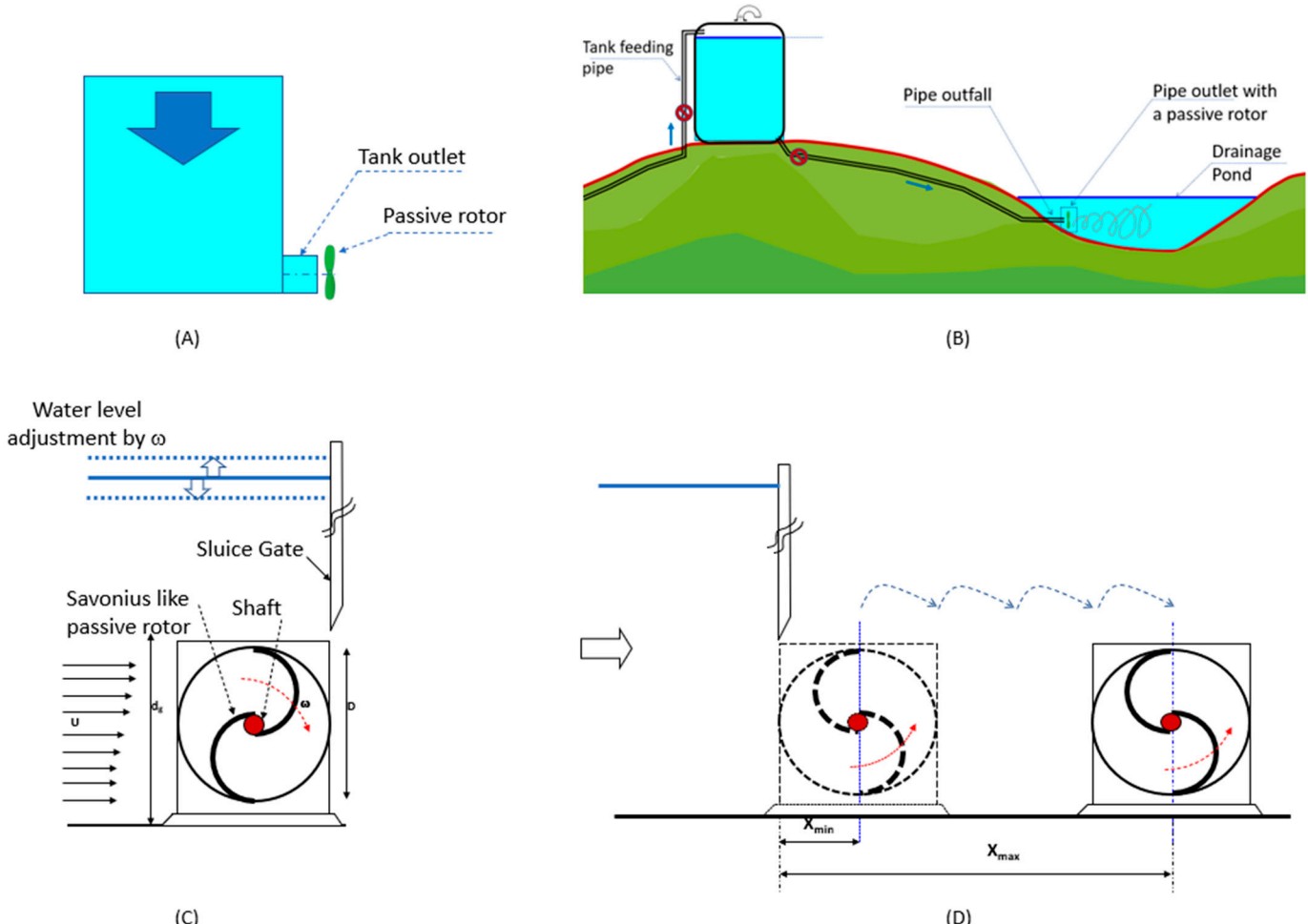

**Figure 1.** Samples of proposed applications where passive rotors could be utilized. (**A**) at the outlet of a drainage tank, (**B**) at the outlet of pipe outfall, (**C**) at the upstream of a sluice gate, (**D**) at the downstream of a sluice gate.

In 2017, a research work [3] studied a water sluice gate's hydraulic performance with a passive Savonius-like rotor. The study showed that adding a passive rotor affects the

upstream water afflux, and the rotation of the rotor could control this effect. The study also proposed replacing the energy dissipater baffle blocks (frequently used in stilling basins to reduce the energy downstream of hydraulic structures) with some loaded rotors to generate a "useful part of energy while dissipating the harmful part of it". In 2021, the effect of using a passive rotor (at the pipe outlet of a draining tank) on the water drainage rate from the tank was experimentally investigated. The study has shown that adding a symmetric, four-blade, passive rotor increased the average water drainage rate from the tank by up to 9.0% due to the formation of a low-pressure region at the pipe outlet caused by the swirl flow induced by the water–rotor reciprocal interactions [5].

Currently, the authors are involved in a research project investigating the feasibility of using passive rotors at the outfalls of water facility plants (application presented in Figure 1b). The project includes some research objectives. One of these objectives is to experimentally and numerically study the mixing characteristics resulting from using the passive rotors in the near field flow zone. The authors experimentally studied the optimum number of blades and the effect of the rotor similarity in their first paper [5].

This study is conducted to answer the following four questions: (1) Based on the available CFD packages, what is the relevant and practical numerical workflow that could be followed to simulate the temporal variation of the angular speed of the passive rotor given an outlet subjected to time-varying effluent? (2) What is the suitable turbulence model(s) that could be used to obtain the most accurate results for such an application? (3) What is the effect of the pipe outlet–rotor gap distance on the rotor's induced angular speed? (4) From a numerical perspective, and based on the developed CFD workflow (in item 1 above) and the relevant turbulent model (in item 2), will the use of a passive rotor at the pipe outlet to improve mixing characteristics be technically feasible?

Before addressing the first question, it should be emphasized that the numerical manipulations for applications involving a passive rotor are considered challenging since the angular speed of the rotor is not known a priori, and it is affected by the mutual interactions between the fluid (the water) and the structure (the rotor) interactions.

The rest of the paper is organized as follows. The research methodology is presented in Section 2, where the problem statement, the experimental study (used for model validation), and the assumptions are described. In Section 3, the 3D CFD numerical model is established using FLUENT, with workflow development, mesh generation, grid discretization, and boundary conditions. In Section 4, the model validation is conducted, where the selected model discretization is examined, the sensitivity of the used turbulence models is assessed, and the required computational resources are discussed. Section 5 discusses the passive rotor's results on the near flow field of the outfall and its corresponding expected effects on the mixing characteristics. Section 6 concludes the findings of the present paper and states the recommendations.

## 2. Method Statement

### 2.1. Study Objectives

The first aim of the current study is to propose a numerical workflow that first helps simulate the variation in time of the angular speed of a passive rotor subjected to time-varied effluent flux. Then, the developed numerical workflow will be used to assess the technical feasibility of adopting the passive rotor element at the pipe outlets to enhance mixing in the near field downstream of the outfalls.

Therefore, the study objectives are to respond to the following questions:

What is the appropriate numerical method to deal with the moving rotor, and what is the practical numerical workflow that could be adopted to simulate the variation in time of the angular speed of the passive rotor given an outlet subjected to time-varying effluent?

What suitable turbulence model(s) could be used to obtain the most accurate results for such an application?

Is it technically feasible to use the passive rotors at the pipe outlets to enhance the mixing characteristics of the near flow field?

## 2.2. Physical Experiment

This subsection briefly presents the physical experiment used to verify the numerical model. More details about the conducted experiments are found in [5]. The experimental setup used to investigate the hydraulic performance of a submerged 18 mm PVC pipe outlet equipped with a passive rotor is presented in Figure 1. The pipe outlet is a drainage facility for a falling head vertical plexiglass tank. The tank's height and diameter are 68 cm and 14 cm, respectively. The tank is placed about 30 cm above the water surface of the mixing tank (tank 2, Figure 1b). The mixing tank has a constant level with horizontal dimensions and a height of 69 cm × 30.5 cm and 28 cm, respectively. This tank is composed of two similar chambers separated by a plexiglass baffle with a height of 24.2 cm (a side weir). The pipe outlet is submerged under the fixed headwater surface in tank No. 2. Water passes from the first to the second chamber of tank No. 2 via the internal side weir. A side circular outlet in the second chamber is used to maintain the water level in the tank.

Figure 1c,d present the side and front views of the pipe outlet with a four-bladed passive rotor. The diameter of the rotor and the distance between the pipe outlet face and the rotor centroid are 31 mm and 13.5 mm, respectively.

Tank 1 was filled with water for the experiment's preparation by opening the submerged pump, valves 1 and 2, while keeping valve 3 closed (Figure 1b).

## 2.3. Measurements via Video Tracking

The lab experiment requires the measurements of two parameters over the course of time. These parameters are the tank water level and the angular speed of the passive rotor. The angular speed can be measured using tachometers and stroboscopes. Recently, the video tracking (VT) approach has been successfully used in many applications [4]. The VT technique has many advantages. For instance, the VT system is relatively inexpensive, as the tracking can even be conducted by the available smart-phone cameras, it can be used to capture the temporal variations of the rotor angular speed, and it can also be used to measure the angular speed for submerged revolving objects (similar to the study on hand). Nevertheless, the VT system has its own limitations, which will be discussed later on in the following sections.

Two digital cameras (cameras 1 and 2) were used to capture the variations of the rotor's angular speed and the water elevation in the falling head tank (tank1), respectively. The first is a bridge-type Fuji Finepix digital camera (Tokyo, Japan), and the second is a Microsoft LifeCam Studio webcam (Redmond, WA, USA).

The freeware video analysis Tracker package (version 6.0.4, from Open Source Physics Project by National Science Foundation, USA) was used to analyze the recorded videos for the rotor and the water surface in tank 1 using the auto-tracking tool. All the angular velocity measurements were conducted at an image capturing rate of 240 fps.

To measure the angular speed using the Tracker package, a 4 mm circular-black object is drawn near the tip of one of the rotor's blades; refer to Figure 2d. The circular object is auto-tracked via the Tracker. Using the tracked coordinates of the center of the circular object, the Tracker could determine the instantaneous variations of the angular speed, as shown in Figure 3. The time-averaged values of the angular speed were later calculated by applying the moving window average technique while considering an average time of one second. The reader can refer to [5] for more details about the tracking manipulation.

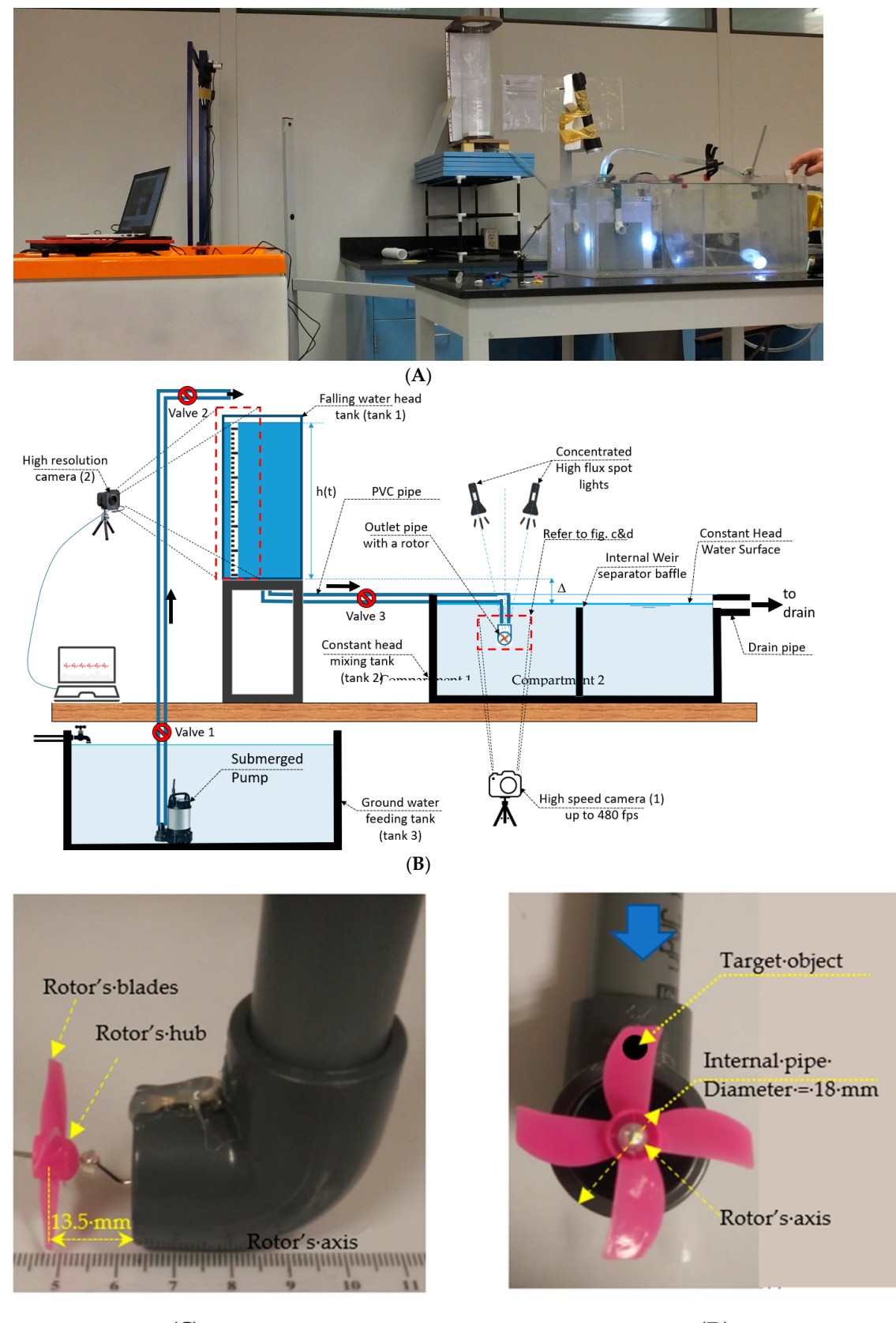

**Figure 2.** Experimental setup: (**A**) a snapshot of the experimental setup, (**B**) Elements of the experimental setup, (**C**) side view of the rotor, (**D**) front view of the rotor.

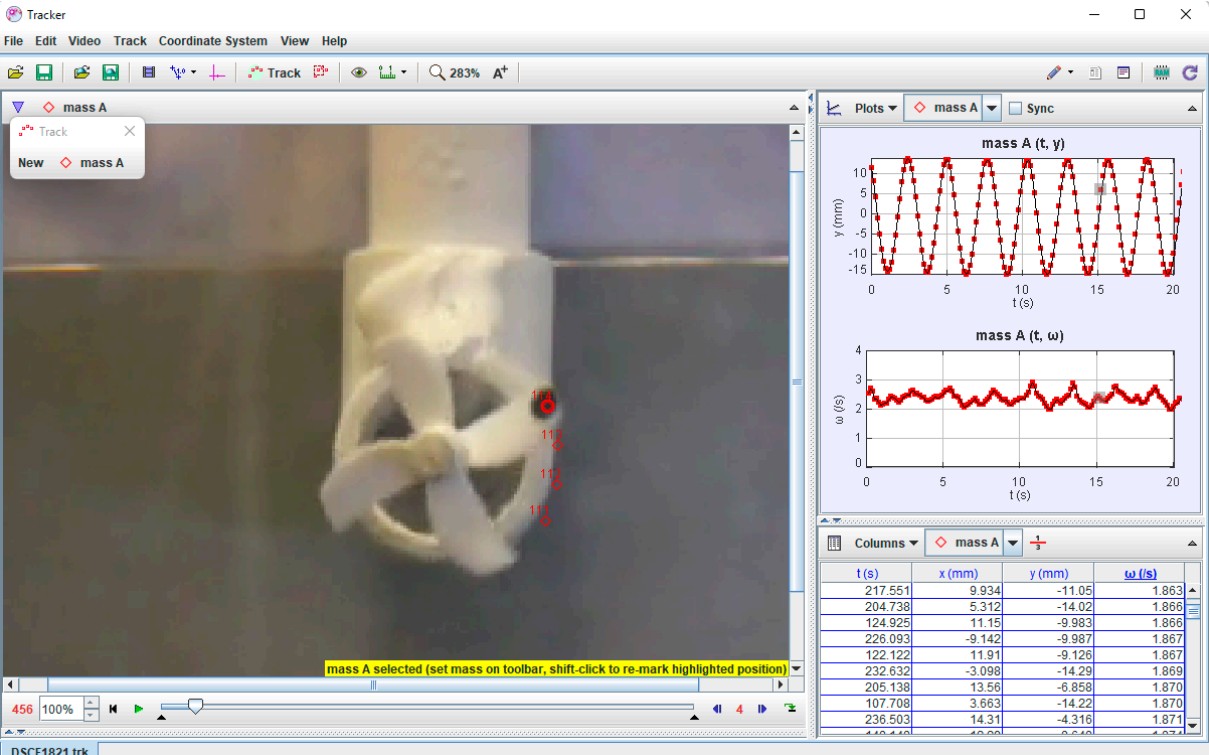

**Figure 3.** Auto-tracking of the rotor's angular speed via the Tracker (camera capturing rate = 240 fps).

### 2.4. Accuracy of Measuring the Angular Speed Using Video Auto-tracking Tracking

Two tracking techniques are commonly used in the Tracker. The first is manual tracking, where the user is obliged to manually identify the location of the center point of the tracked "target" object in each frame of the video recorded during the measurement. The second technique is auto-tracking, where the location of the tracked object is automatically identified by the software. Due to the large number of image frames (240 times the duration of the recorded video in seconds) that need to be processed in each video measurement, auto-tracking is adopted in this study. The auto-tracking is based on creating one template image of a selected feature of interest (target object) and then searching each frame for the best match for that template. The best match is the one with the highest match score.

Despite that, auto-tracking, on the one hand, is more convenient and faster and consumes less time than manual tracking. On the other hand, it could result in less accurate measurements. The expected reduction in accuracy due to the auto-tracking is due to what is called template evolving and drifting, in which "visual" or "apparent" changes might take place regarding the shape and or the colors of the target object (for instance, due to the changes in the light intensity). These changes could lead to a small drifting or small errors in the "looking for" coordinates of the center of the tracked object. This means that using auto-tracking to measure the angular speed of a rotor could result in expected small induced or pseudo-perturbations in the angular velocity that are not physically existing in reality. Nevertheless, such small errors are not expected to affect the time-averaged values of the angular speed measurements due to the expected randomness of their occurrence.

To eliminate such errors, a time averaging process (using the marching window technique) is commonly used and applied to all the measurements based on a duration interval of 1 s.

To assess the accuracy of the time-averaged measurements of the angular speed of the rotor using visual tracking, a number of diagnostic experiments (rotor speed from 160 to 1500 rpm) have been conducted to measure the rotor angular speed using two different measurement techniques. The first technique is the digital photo tachometer, and the

second is the visual (video) tracking. These experiments are conducted, where the rotor is kept running in the air (not submerged under water) to avoid the dispersion of the reflected laser beam (in the digital photo tachometer) that is expected to take place when the system is set underwater.

The tested rotor is attached to the axis of an overhead stirrer (Dragon Lab Type model OS40 Pro, Dragon Lab, Rowland St. City of industry, CA, USA). A small piece of reflective tape was stuck to the tip of one of the blades of the rotor, and a handheld, non-contact digital tachometer (model: NEIKO 20713A, NEIKO, Wilmington, Delaware, USA) was used to carry out the non-contact diagnostics. The measurements of the tachometer are based on identifying how many times per minute the laser beams (that emerge from the tachometer) are reflected back (from the reflecting tape that is stuck on the surface of the rotor) to the tachometer. Figure 4 shows the setup of this assessment experiment.

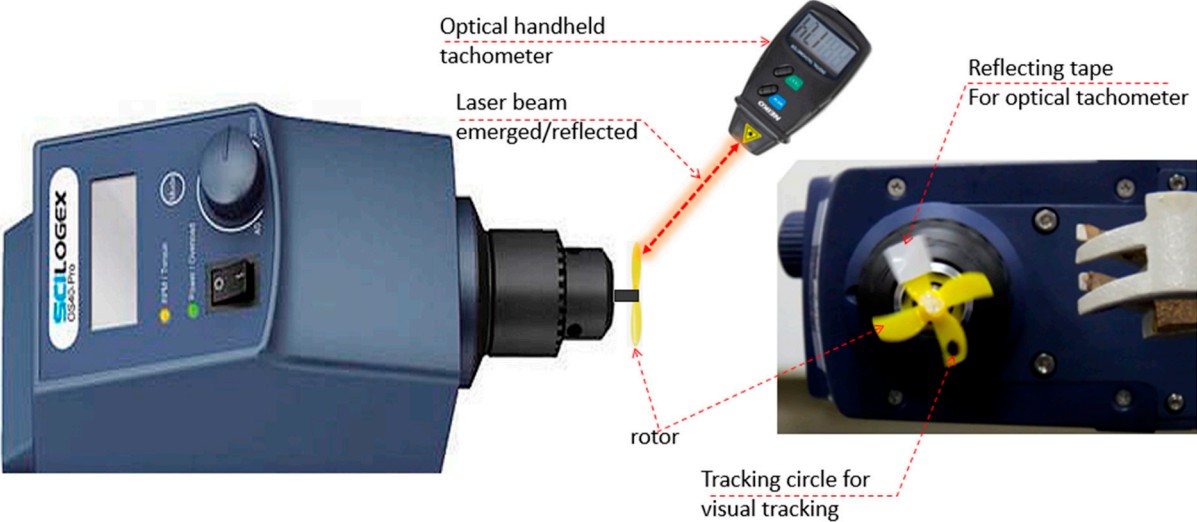

**Figure 4.** Setup of the assessment of accuracy experiments (using two different methods) for the angular speed measurements: oblique-plan view, side view.

The accuracy of the tachometer is of the order of ±0.05%, and the discrepancies between the measurements from the two techniques were found, in general, to be of the order of +1%.

*2.5. Numerical Techniques for Rotating Elements in FLUENT*

FLUENT provides three different methods to deal with rotating objects. These methods include the moving reference frames (MRF), the sliding mesh (SM), and the six-degrees-of-freedom (6DOF) [6]. Accordingly, it will be essential to identify the pros and cons of each method, to decide what is the most suitable calculation method for the current application, and to determine the practical numerical workflow that is to be adopted to carry out accurate predictions of the time-varied rotor angular speed subjected to the available computational resources.

The CFD literature reveals that the moving reference frame (MRF) method can be applied for the steady-state simulation, where the rotation speed of the rotor can be considered a constant [7–10]. It should be noted that the MRF has a low computational cost and can accurately predict the characteristics of the constant rotation applications.

In the case of unsteady-state flow, if the rotor's angular speed $\omega$ is known as a function in time, then the sliding mesh "SM" technique may be recommended. It should be noted that the sliding mesh technique has been widely implemented to simulate the rotor motion in many turbine applications [11–14]. Nevertheless, it should be mentioned that the computational time resources needed for the sliding mesh technique for unsteady applications are usually 30 to 50 times higher than the steady-state applications [15]. In

recent years, an alternative technique known as the overset (chimera) mesh method has caught the attention of the academic community. A comparison between the overset and the sliding mesh methods is used to find their advantages and disadvantages in the two-dimensional simulation of vertical axis turbines [16]. The comparison was established to predict the performance parameters of the turbine in order to check the capabilities of the models to capture the complex flow phenomena in these devices and the computational costs.

The dynamic mesh (6DOF) method is appropriate when the rotor's angular speed as a function of time is not known beforehand; therefore, the subsequent move of the rotor is determined based on the solution at the current time. In this case, the rotor's angular speed $\omega$ is computed from the force balance on a solid body via the 6DOF solver. The update of the volume mesh is handled automatically by FLUENT at each time step based on the new positions of the boundaries. The starting volume mesh and a description of the motion of any moving zones in the model have to be provided to use the dynamic mesh technique. It should be noted that the 6DOF method has been used to simulate the rotation of the rotor in different research works, such as [17–20].

Research work [20] compared the experimental results with the results of the CFD model while using the 6DOF technique at a pico scale. They found that the 6DOF method is more accurate than the MS method for predicting the performance of cross-flow turbines. However, the moving mesh requires less computational time and a faster convergence rate than the 6DOF.

In [21], the researchers used a glyph script in POINTWISE-GRIDGEN to generate the domain and mesh of a vertical-axis marine (Water) turbine (straight-bladed Darrieus type), with particular emphasis on the turbine's unsteady behavior. At the same time, the simulations were performed in ANSYS FLUENT v14 (Ansys, San Jose, CA, USA). To simulate the interaction between the dynamics of the flow and the Rigid Body Dynamics (RBD) of the turbine, a User Defined Function (UDF) was generated for the quasi-steady-state.

Research work [5] investigated the effect of adding a passive rotor on the outlet performance, where different sizes and numbers of blades of rotors were considered. In the current research, the CFD simulations have been applied to study the $\omega$ of a passive rotor starting from a stationary state under a decelerated cross submersible jet for one size and one shape of a rotor. From the physical model [5], the current numerical study assumes the rotor's recommended size and shape.

In this research, the rotor is assumed to be submersible, not near the water surface; therefore, there is no need to include the free surface and two-phase problem (water and air), such as in [22] and [23]. Moreover, the rotor is also not near the bed, and, thus, the effect of the sediment will not be included [24]. Additionally, the flow is gradually decreased, not rapidly varied, as in [25].

Starting the rotor rotation from a stationary state to reach a constant speed under steady cross velocity has rarely been tackled by researchers, as it has in wind turbines [26]. To the best of the authors' knowledge, the case of a rotor starting to move from a static state under a decelerated cross-water velocity is studied for the first time. This case study is more complicated since $\omega$ never achieves a steady state. The rotor's angular velocity rapidly accelerates for a very short time; then, it gradually decelerates. Therefore, the 6DOF technique has been applied, since no known rotating speed is unknown to the computational grid frame. The 6DOF model has three objectives: the determination of $\omega_{max}$, the variation of $\omega$ with time, and the equation of $\omega$ as a function of $v$. The 6DOF results were compared with those obtained from the physical model [5]. The verification includes both $\omega_{max}$ and the variation of $\omega$ with time. After that, the $\omega$ equation as a function of $v$ was investigated. Then, the sliding mesh technique was used after adding that equation to the User-Defined-Functions (UDF). The accuracy of both CFD methods and the time consumed are discussed. Finally, the results and a discussion of the sliding mesh model are tackled.

*2.6. Assumptions and Simplifications*

The numerical study on hand considers the following main assumptions:

- There is no eccentricity for the shaft of the rotor to the center of the pipe outlet;
- The shaft of the rotor revolves smoothly around its pivot without resistance;
- The shaft and its rotor have the same angular speed (i.e., no slipping took place);
- No deformation is allowed regarding the blades of the rotor during the simulation;
- The rotor blade is assumed rigid enough, and its deformation is neglected;
- The direction of the water flux from the pipe outlet (before impinging on the rotor) is mainly horizontal, with no inclination;
- For practical reasons, the main focus of the simulation is directed on capturing the decelerated period for the rotor (refer to the calibration section) since this period is the dominating stage, and the accelerating zone lasts less than 1.5% of the whole simulation time.

## 3. Numerical Model Development

*3.1. Workflow Development*

Since the rotor is subjected to decelerating water flux and its angular speed is not known a priori and never reaches a fixed value, the dynamic mesh method seems to be the suitable technique to determine the temporal variations in the rotor's angular speed. The downside of adopting the 6DOF technique is that it requires substantial computational and time resources. As a compromise, it is proposed to adopt the following computational workflow:

- Start the simulation using the dynamic mesh method using the 6DOF solver for about 21% of the required total simulation time (for the study on hand, for the first 40 s);
- The k-e turbulence model will be initially selected, and the accuracy of the generated mesh will be checked near the boundaries by assessing the y+ plot;
- The simulation will be repeated many times, and each time, a different turbulence model will be tried, and the rotor's angular speed results will be compared with the measurements;
- Identify the most relevant turbulence model that gives the best match with the measurements;
- Based on the obtained results of the optimal turbulence model, identify the relation between the rotor's angular speed ($\omega$) and the pipe outlet velocity ($\upsilon$) and create a user-defined function (UDF) for it;
- Switch the model to the sliding mesh (SM) technique while adopting the optimal turbulence model, and start the simulation until the end of the simulation time.

Figure 5 presents the block diagram for the abovementioned workflow.

*3.2. The Geometry of the CFD Model*

The assumed passive rotor is typical of the rotor's recommended size and shape for the previous study [5]. The rotor of four blades that is 31 mm in diameter has been developed in three-dimensional space using the workbench in ANSYS FLUENT v14. The tank's geometry in the CFD model is typical of compartment number 1 in tank number 2 in the experimental study, as shown in Figure 6a. The tank in the CFD model, shown in Figure 6a,b, is 30.5 cm in length and 34.0 cm in width, and the controlled water depth equals 25 cm. The sidewall is 25 m from three sides, and the fourth wall (the baffle wall) is 24.2 cm. The water outlet is located above the baffle wall. The rotor axis is fixed at the center of the inlet pipe. The mesh consists of around 4.7 million tetrahedral cells; the cells on the rotor surface are shown in Figure 7a. Figure 7b illustrates the cells on the surface of one blade and the cells on the surface of the rotor, the inlet, and the fixed walls in Figure 7c. The minimum and the maximum surface areas on the rotor are $2.775 \times 10^{-3}$ and $1.086 \times 10^{-2}$ mm$^2$, with an average of $6.716 \times 10^{-2}$ mm$^2$.

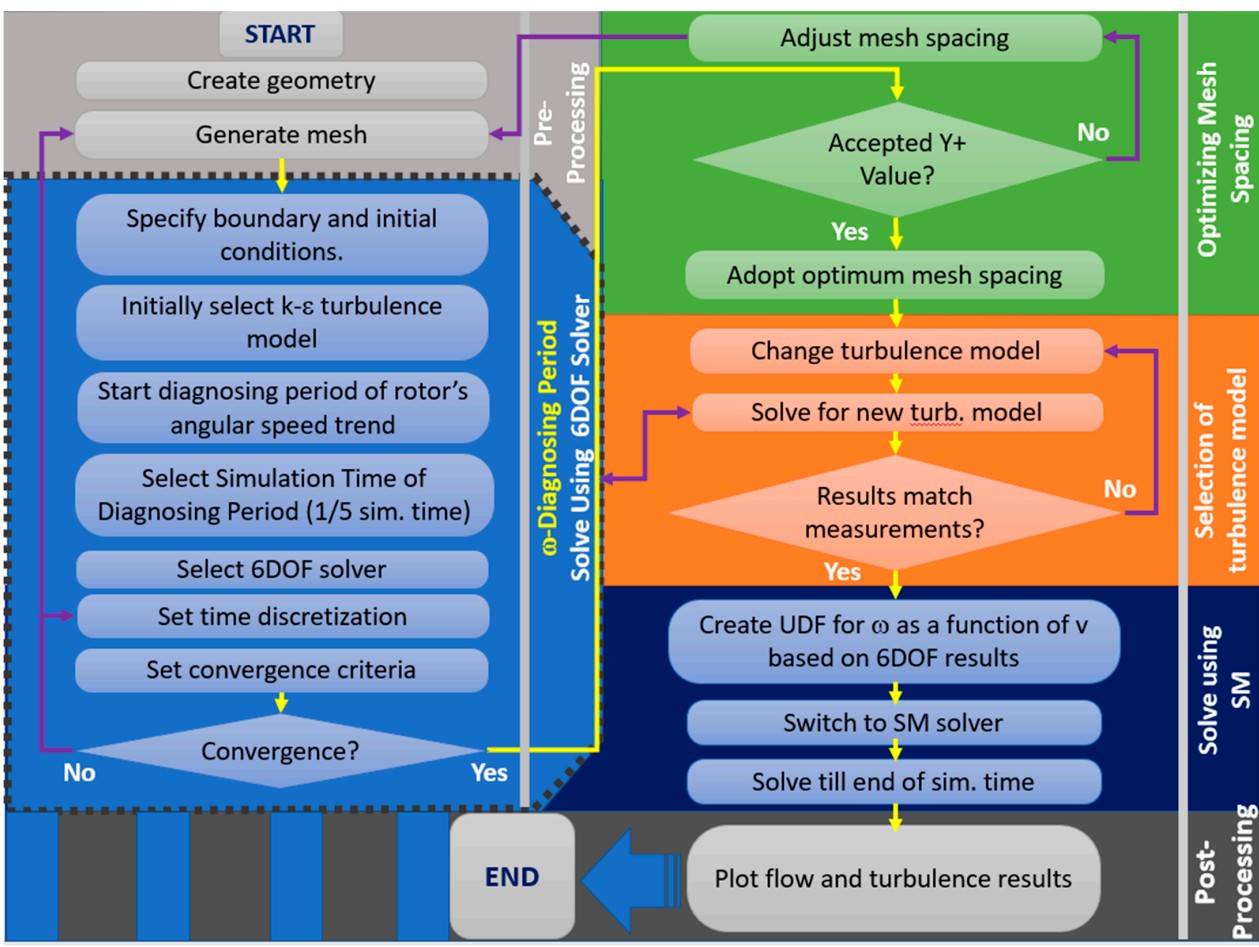

**Figure 5.** Block diagram for the adopted CFD workflow.

### 3.2.1. The Boundary Conditions

The inlet boundary is assumed as a mass inlet boundary, and the outlet boundary is considered a pressure outlet. All other boundaries are taken as walls. The inlet boundary and the solver parameters are discussed hereunder.

#### The Inlet Boundary

Since the inlet velocity in the physical model depends on the falling water head, the CFD's inlet boundary has to precisely simulate the velocity value. The diameter of the inlet is 18 mm at a level of 12.1 cm and is located in the middle of the sidewall, as shown in Figure 6c,d. In the CFD model, the mass flow inlet boundary has been used. The best-fitting curve has been investigated from the experimental results of the flow rate with time. Equation (1) has been assumed using UDF at the inlet. The correlation of the equation is 0.9993.

$$Q = 0.0851 - 0.0002 \times t \tag{1}$$

where:

Q is the mass flowrate (kg/s)
t is the time (s)
This equation is valid only if t < 180 s

#### The Outlet Boundary

The top of the baffle (side) wall is assumed to have an outlet pressure boundary to match the physical model, as shown in Figure 6c,d. The height of the boundary is 8 mm, and the width is 30.5 cm.

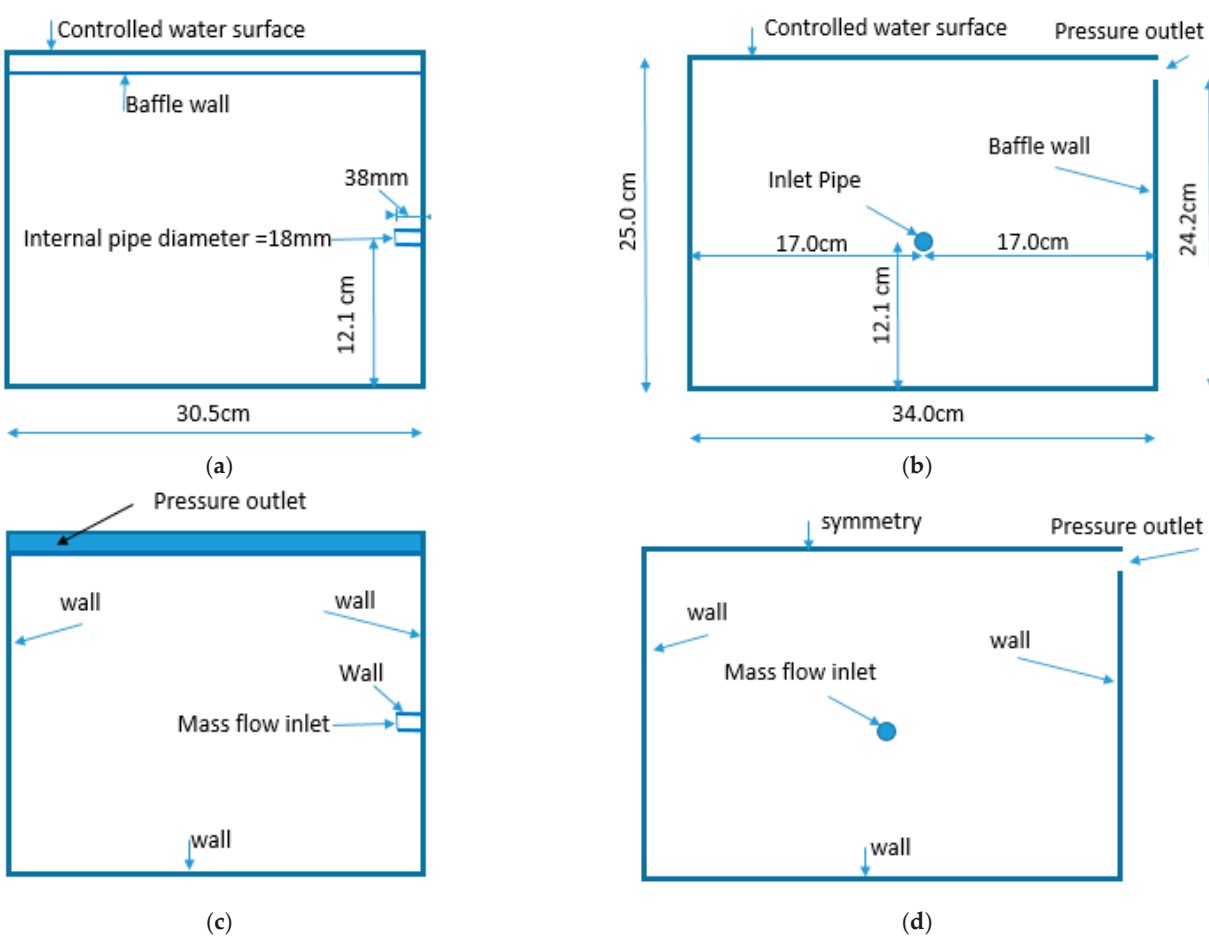

**Figure 6.** Schematic Figure of the CFD model: (**a**) Side view of the CFD model dimension, (**b**) Front view of the CFD model dimension, (**c**) Side view of the CFD model boundary, (**d**) Front view of the CFD model boundary.

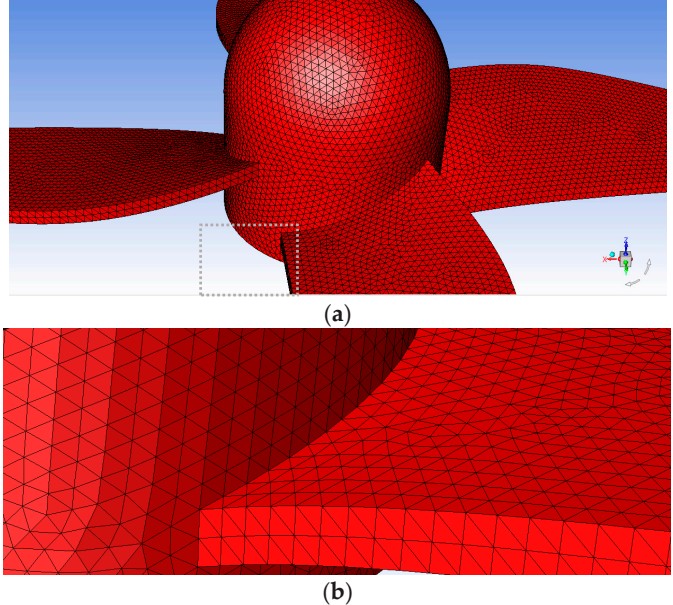

**Figure 7.** *Cont.*

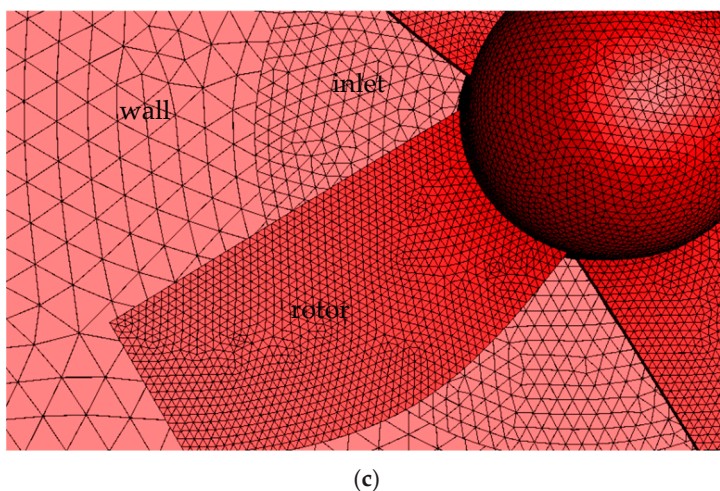

(**c**)

**Figure 7.** Passive rotor grid model: (**a**) The mesh on the rotor surface, (**b**) The detail of the mesh on one of the blades (dotted line zone in Figure 5a), (**c**) The detail of the mesh of the rotor, the inlet, and the walls.

The Top of the Model and the Other Boundaries

The model's top is assumed to be the symmetry boundary, as shown in Figure 6d. All other boundaries (the rotor, the tank sidewalls, and the tank bottom) are considered as walls, as shown in Figure 6c,d.

3.2.2. Solver Parameters

The solver parameter settings for the propeller open water simulations were made with a judicious combination of the FLUENT literature recommendations and the trial-and-error evaluations of various solver settings. Combining these options with the domain dependence and grid dependence studies is a vital, though time-consuming, initial effort before any propeller geometry can be investigated for performance. The final solver parameters, including physical constants, are shown below in Table 1.

**Table 1.** Summary of the boundary conditions and the solver parameters of the CFD models.

| Parameters | Settings |
| --- | --- |
| Solver | Pressure-based, transient |
| Velocity formulation | Absolute |
| Turbulence model | Standard k-ε |
| Water density | 998.2 kg/m$^3$ |
| Water viscosity | 0.001003 kg/m·s |
| Pressure discretization | Body Force Weighted |
| Gravity | 9.81 m/s$^2$ |
| The inlet | Unsteady mass flow inlet |
| The outlet | Pressure outlet |
| The top of the tank | Symmetry |
| All other boundaries | Wall |

**4. Calibration and Accuracy Assessment of the Numerical Model**

*4.1. Measurements of the Rotor's Angular Speed*

Figure 8a shows the measurements of the variation in time of the ensemble average of the angular speed ($\omega$) of the rotor (normalized by the maximum angular speed value $\omega_{max}$). Three distinct stages appear in Figure 8a. The first zone is the acceleration stage (from A to B), where the rotor speed accelerates to achieve its maximum angular speed. Interestingly, the rotor is accelerating at this stage, even though the pipe flow is decelerating. In the second stage (from B to C), the angular speed starts linearly and progressively decreases

with time until it reaches point C, where the tank becomes wholly drained. In the last stage (from C to D), the transmission pipe rapidly drains until it reaches point D. This rapid decrease in angular speed may be explained by the substantial difference between the tank and pipe cross-section areas.

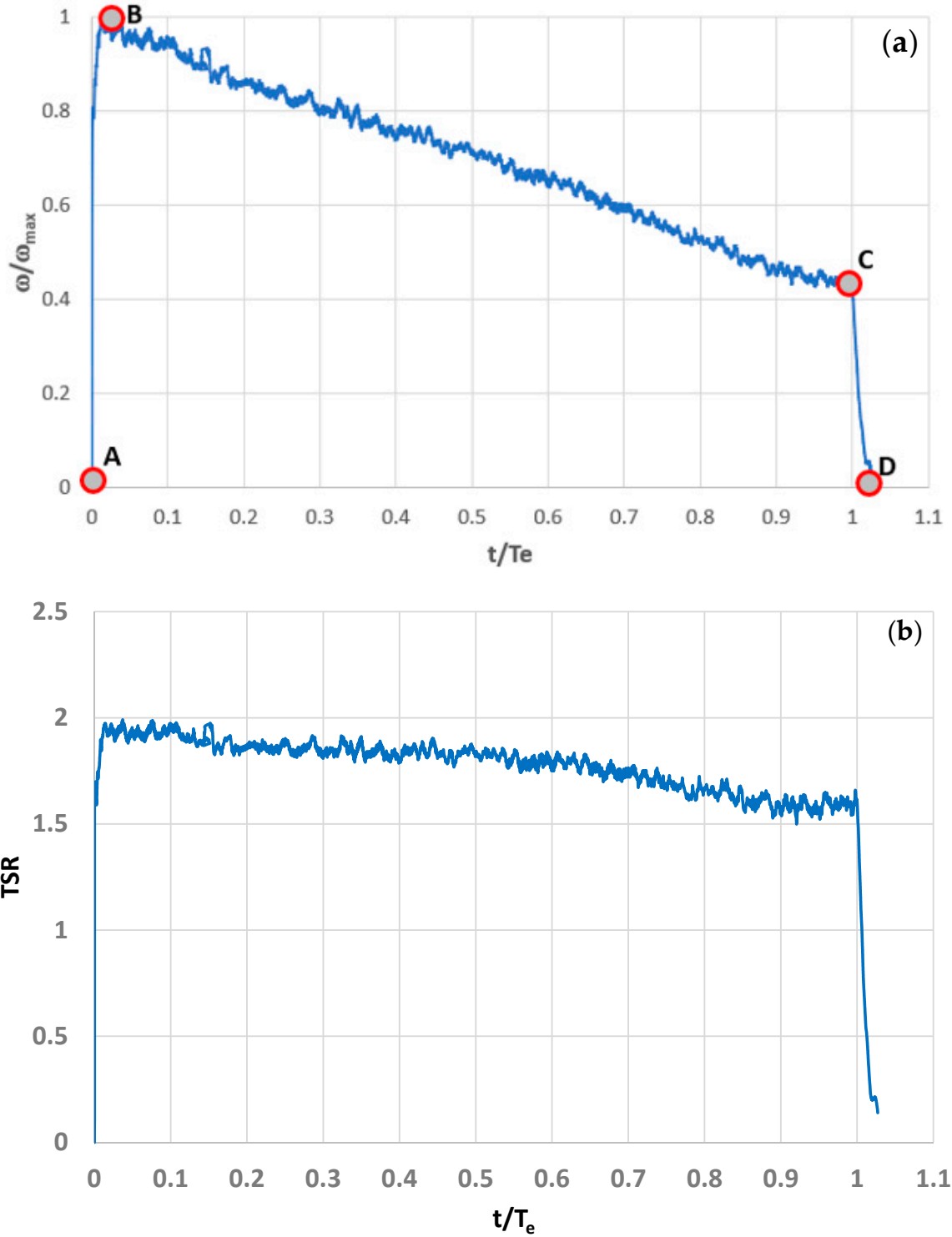

**Figure 8.** Video analysis measurements of the temporal variations of (**a**) the normalized rotor's angular speed and (**b**) the tip speed ratio (TSR).

The rotor's tip speed ratio (TSR) is expressed using Equation (2)

$$TSR = \frac{V_r}{V_o} \qquad (2)$$

where:

The rotor's tip speed could be calculated as $V_r = \omega.d_r/2$, and Vo is the water pipe outlet mean velocity. Figure 8b shows the variations in the rotor's tip speed rotation (TSR) over the course of time. The figure shows that TSR varies from about 1.6 to 1.9, which means that the tip rotor speed is almost 60 to 90% faster than the outlet water velocity.

### 4.2. Model Discretization and GCI Analysis

Numerical models usually use discrete methods to convert the governing partial differential equations into algebraic equations. All discrete methods introduce discretization errors that might be significant enough to ruin the accuracy of the produced numerical solutions. Therefore, the computational fluid dynamics community and many other CFD-reputable journals require discretization error estimation as a prerequisite for publishing any CFD paper.

The mesh discretization has been studied by applying the grid convergence index (GCI). The generated mesh was assessed by calculating the non-dimensional wall distance for a wall-bounded flow (y+). The validity of the turbulence models is examined by comparing the models' results with the experimental measurements of the rotor's angular speed.

The Grid Convergence Index (GCI) is a relatively new discretization error estimation technique. The GCI is calculated to answer whether the adopted mesh in the simulation is refined enough or not. A minimum of two mesh solutions are required, but three are recommended to calculate the GCI.

In the current study, three groups of mesh sizes with different grid densities are established and are named Low, Med, and High grid density, respectively. The corresponding numbers of mesh cells are almost 1.0, 2.7, and 4.7 million. The three groups of mesh sizes were used to calculate the rotor's maximum angular speed and compare it with the measurements. The GCI was calculated using the method described in [27]. The GCI for the high grid density group was less than 0.8%, and the maximum rotor angular speed error was less than 1.4%. Therefore, the mesh of 4.7 million cells was selected for further analysis.

### 4.3. Checking Model Discretization near the Boundary

To verify the numerical accuracy near the boundaries, the non-dimensional wall distance for a wall-bounded flow (y+) is calculated for the first layer of the generated grid points throughout the rotor at the instance of maximum angular speed (refer to Figure 9a). The rotor center is presented by the zero point in the figure, as shown in the key figure. Two blades (r = 15.5 mm) are almost horizontal, and the other two are almost vertical. Figure 9b shows the 3-D variation of y+ along the transect A-A. It is noted that the y+ values lie within the acceptable limit of 5 (the maximum Y+ is less than 5 for the rotor, with an average value equal to 2.34, while the maximum Y+ is less than 1 for all other fixed walls, with an average value equal to 0.6 [28,29]).

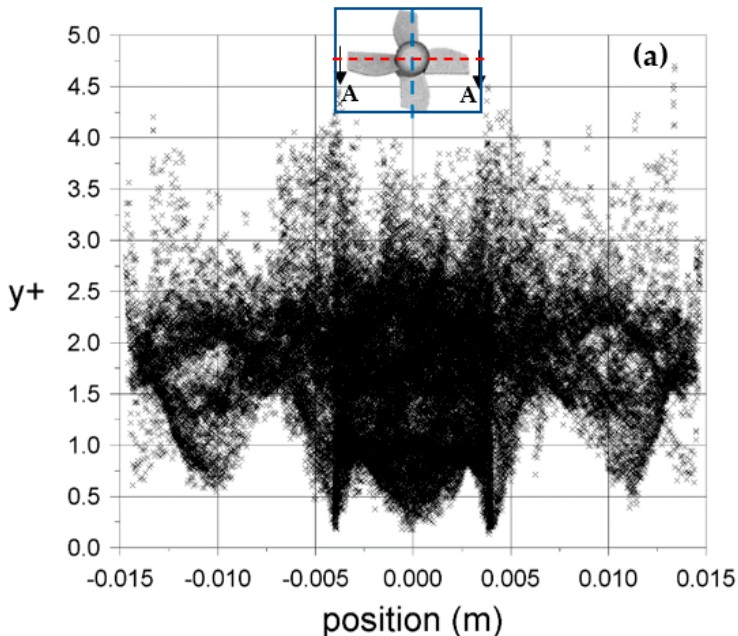

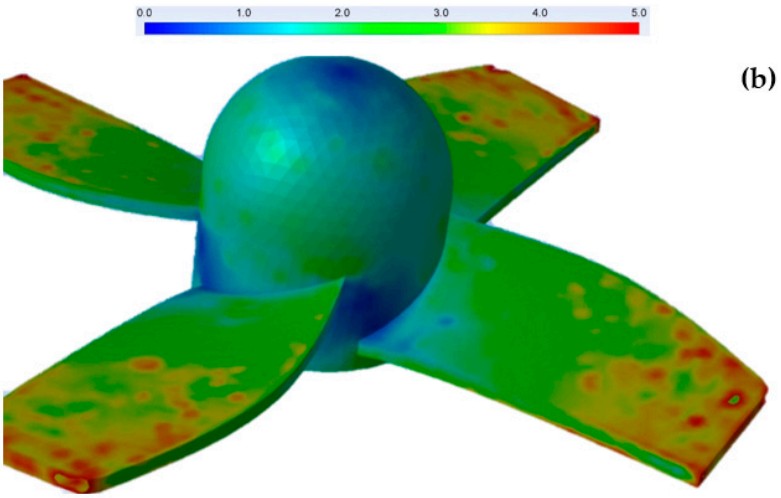

**Figure 9.** The relationship between the position on the blades and the y+ value at the maximum angular velocity. (**a**) Spatial variations of the wall y+ value along transect A-A, (**b**) Color contours of y+ over the whole rotor.

*4.4. The Sensitivity of the Turbulence Model*

The validation of the turbulence model is of fundamental importance for evaluating the reliability of the CFD simulations. Several research studies have used FLENUT to study the rotor motion for different applications, where different turbulence models have been adopted. Since the current study topic has not been tackled before, to the best of the researchers' knowledge, the sensitivity of the turbulence models should be studied to select the most appropriate turbulence model with a maximum speed of 0.6 m/s. Almost all of the different turbulence models applied earlier in other research on rotor movement are considered in this study. Eight different turbulence models have been analyzed in this research. A list of these turbulence models considered in the sensitivity analysis is given below:

- Spalart–Allmaras model [30,31],
- The K-ε standard model, [32–35],

- The K-ε RNG model, [36],
- The K-ε realizable model [31,37],
- The K-ω standard model [38],
- The K-ω-SST model [31,39,40],
- The Reynolds Stress Model (RSM) [41],
- The Transition SST (four equations) [18]

The sensitivity study was analyzed in two steps. First, the different turbulence models are compared in terms of how they predict the maximum angular speed of the rotor. The turbulence model that provides the most accurate value of the maximum angular speed is selected. The second step was used to check the accuracy of the chosen turbulence model by comparing the reduction rate of the angular speed with time to the physical model

### 4.4.1. The Maximum Angular Speed

The maximum rotation speed results for the different turbulence models have been compared directly to those obtained in the physical model. The results are shown in Figure 10a. Figure 10b shows the percentage error in the predicted maximum angular speed for each turbulence model. It is interesting to notice that the three K-ε models yield, in general, the most accurate results compared with the other turbulence models. It is found that the standard K-ε model produces the lowest error (2% error), which is slightly better than the other K-ε models. Therefore, the standard K-ε model has been selected to proceed to the next validation step. It is also of interest to notice that the K-ω standard turbulence model resulted in the most significant error (27% error) among the other models

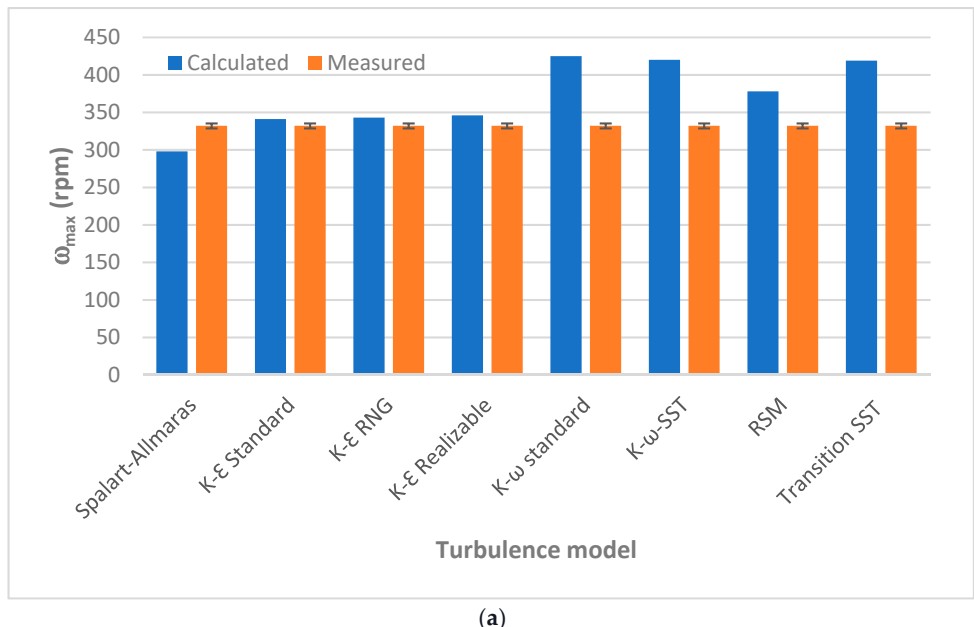

(**a**)

**Figure 10.** *Cont.*

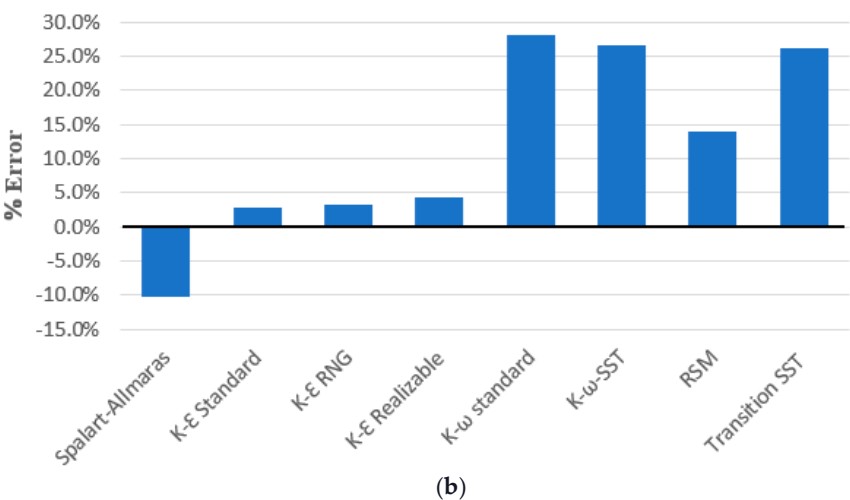

(**b**)

**Figure 10.** Predicted rotor's angular speed based on different turbulence models. (**a**) maximum angular speed, (**b**) percentage error in the maximum angular speed.

### 4.4.2. Comparison of the Temporal Variation in the Rotor's Angular Speed

After selecting the standard K-ε model as the optimal turbulence model for this study (maximum speed = 0.6 m/s), a more comprehensive analysis has been applied by comparing the temporal variation in the rotor's angular speed for both the experimental and 6DOF models. Figure 8a illustrates the time decline trend of the rotor's angular speed for the first 40 s. In general, the measurements and the CFD-6DOF model match reasonably well. Nevertheless, the decline rate of the angular speed of the rotor predicted by the CFD-6DOF model is slightly steeper than the measurements. Some discrepancies exist between the 6DOF model and the measurements, especially within the accelerating phase (from points A to B in Figure 11a,b). Such discrepancies are probably related to the non-homogeneous mechanical resistance experienced by the rotor's shaft in the physical model while initiating the rotor's movement. After the rotor has gained its angular velocity, both trends of the curves are almost identical. Overall, the results of the 6DOF model are very consistent with the experimental results.

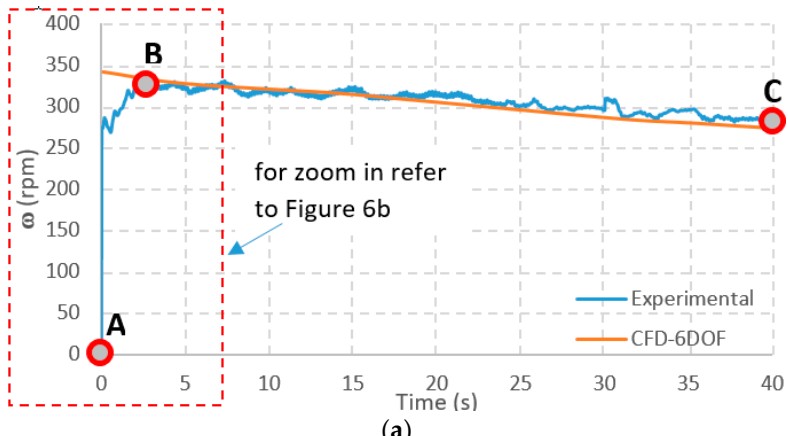

(**a**)

**Figure 11.** *Cont.*

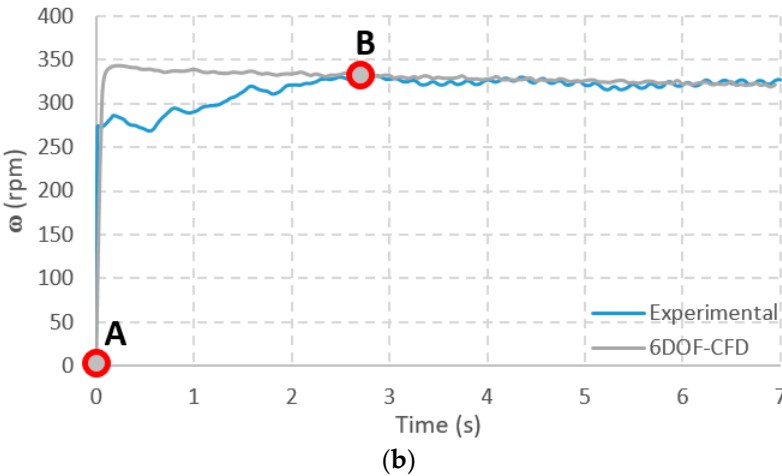

(**b**)

**Figure 11.** The variation in the rotor's angular speed: (**a**) The time span from rest to 40 s, (**b**) Zoom in for the accelerating zone. (for explanation of points A–C, refer to Figure 8a).

*4.5. Computational Resources*

The entire CFD study lasted around three months, with a processing speed of 3.1 GHz, 32 used cores, and 15,000 used core-hours.

Using the 6DOF solver for such a case yields accurate results but consumes much time. A total of 11,500 core-hours are required to conduct only forty seconds for the passive rotor with 4.7 million tetrahedral cells. Only forty seconds showed in the 6DOF simulation (out of 180 s, which was the total time for the experimental run), which was sufficient to deduce the mathematical relation between the rotor's angular speed ($\omega$) and the average water velocity at the pipe outlet ($\upsilon$). This equation has been included in the new UDF for the next sliding mesh simulation step.

*4.6. The Relation between the Angular Speed and the Pipe Outlet Water Velocity*

The relation between the angular velocity "$\omega$" and the average pipe outlet water velocity "$\upsilon$" is linear, with a correlation of 0.998, and the mathematical relationship was deduced from the results of the 6DOF simulation. It was given by Equation (3) below.

$$\omega = 12.0425\,\upsilon - 67.785 \tag{3}$$

where

$\omega$ is the angular speed of the passive rotor (rpm)

$\upsilon$ is the average pipe outlet water velocity (cm/s)

The equation is only valid for the gradual decelerating zone (from B to C in Figure 10a), where $\upsilon$ lies within the following range (19 cm/s < $\upsilon$ < 35 cm/s).

**5. Results and Discussion**

*5.1. Perturbation of Angular Speed*

The sliding mesh model, which is a particular case of general dynamic mesh motion wherein the nodes move rigidly in a given dynamic mesh zone, is applied to the case study. Additionally, multiple cell zones are connected through non-conformal interfaces. As the mesh motion is updated in time, the non-conformal interfaces are likewise updated to reflect the new positions of each zone [42,43]. It is essential to note that the mesh motion has been defined using the investigated Equation (3). The model has been used with the same setup, except with the sliding mesh instead of the 6DOF. Additionally, the UDF defines two equations: the relation between the time and the velocity, as stated in Equation (1), and the relationship between $\upsilon$ and $\omega$, as shown in Equation (3). Figure 12 represents the relationship between the angular speed (rpm) and the time (s) for the experimental

and sliding mesh-CFD models along the physical test time (180 s). The time consumed is significantly reduced compared to the 6DOF.

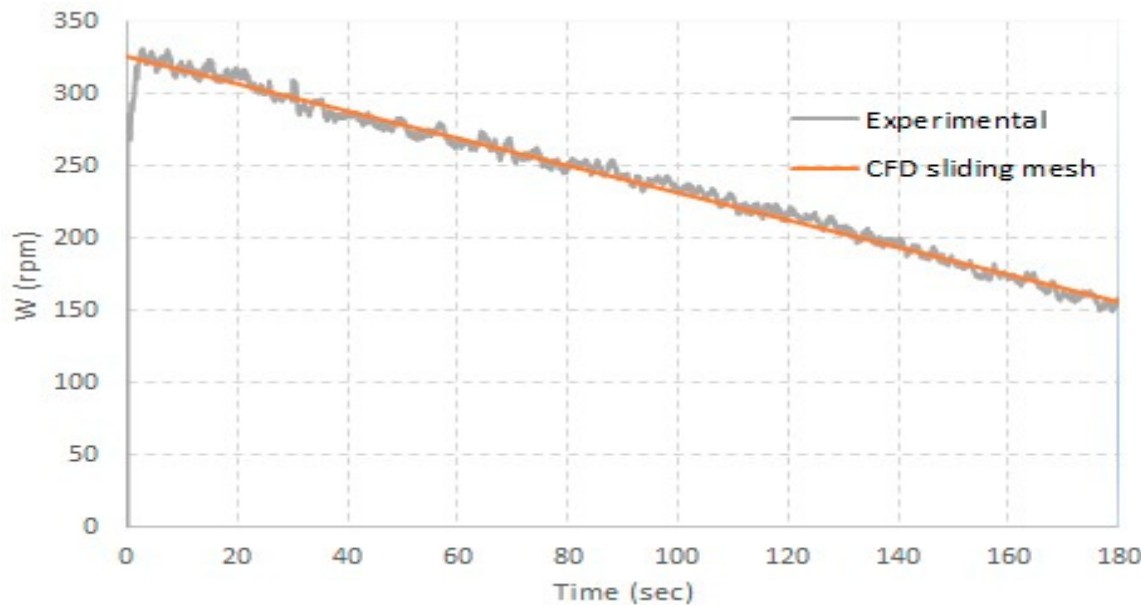

**Figure 12.** The relationship between the angular velocity (rpm) and the time (s) for the experimental and sliding mesh-CFD models.

Interestingly, both the experimental values and the results of the 6DOF model presented some perturbations for the rotor's angular speed with different amplitudes, as shown in Figure 10b. On the other hand, no perturbations exist in the results of the sliding mesh model, as shown in Figure 12, since the model is based on a given mathematical relation for the angular speed. Nevertheless, the perturbation range of $\omega$ in the physical model is more noticeable and considerable than that in the 6DOF CFD model. Such differences could be due to the assumptions stated in the numerical model, which might not 100% comply with the physical situation. Still, the average angular speed values with time for the physical and numerical results are almost identical.

*5.2. Effect of the Outlet–Rotor Gap Distance*

It is of interest to study the effect of the gap distance (s) between the pipe outlet and the passive rotor on the gained angular speed of the passive rotor. For this, five numerical runs were conducted using the 6DOF model while adopting different pipe outlet–rotor gap distances. Figure 13a shows the temporal variations in the normalized angular speed of the passive rotor for different gap distance ratios. Unsurprisingly, it is observed that as the distance between the rotor and the outlet increases, the amount of emerged jet energy that is received by the rotor decreases, and, therefore, the rotor's maximum angular speed decreases as well. As a result, the rotor's maximum angular speed takes longer to attain, resulting in a longer initial acceleration phase (refer to the dashed red arrow in Figure 13a). The power-law formula fits the reduction trend of the maximum rotor's angular speed with the rotor–pipe outlet gap distance ratio well, as depicted in Figure 13b.

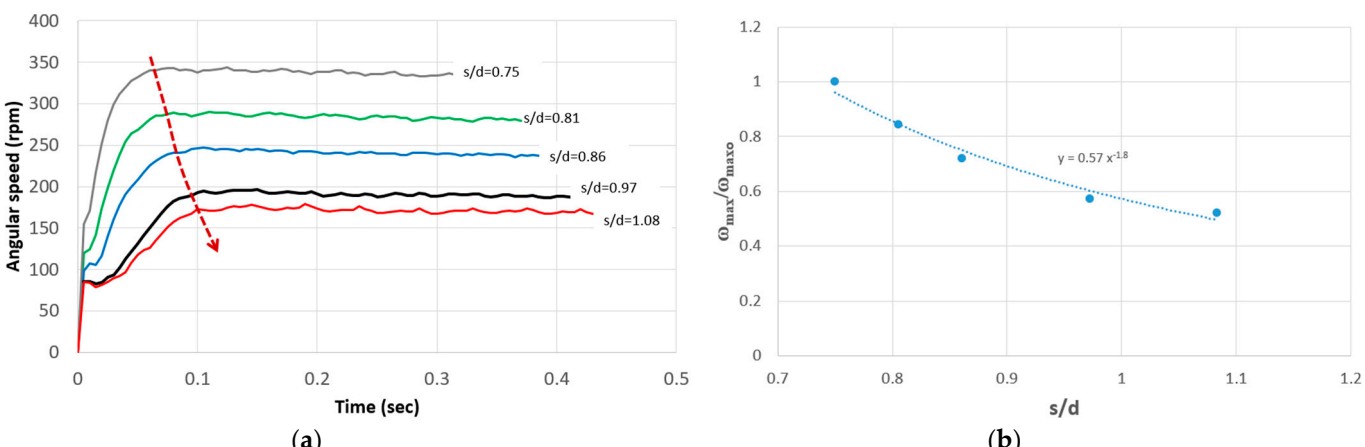

(**a**)                                                    (**b**)

**Figure 13.** Effect of the pipe outlet-rotor gap distance on the gained rotor's angular speed: (**a**) Temporal variation with time, (**b**) Maximum rotor's angular speed.

### 5.3. Effect of the Passive Rotor on the near Downstream Flow Field

Figure 14 shows an example of the general 3D flow visualization of the flow path-lines downstream of the passive rotor (for the case of $\upsilon_{av}$ = 30 cm/s and $\omega$ = 293.5 rpm). The path lines are also color-coded based on the corresponding values of flow velocity, where a relatively high flow velocity exists near the rotor (within a distance of 2–3 d, where d is the pipe outlet diameter), and the flow velocity significantly decreases downstream. The figure illustrates how a spiral flow is developed by the passive rotor when a rotating wake occupies a flow domain whose average lateral size exceeds the rotor's diameter ($\approx$2.3 D, where D is the rotor's diameter). The rotating wake influence is also observed to extend downward to the front wall of the tank (x $\approx$ 15 d, where d is the pipe outlet diameter).

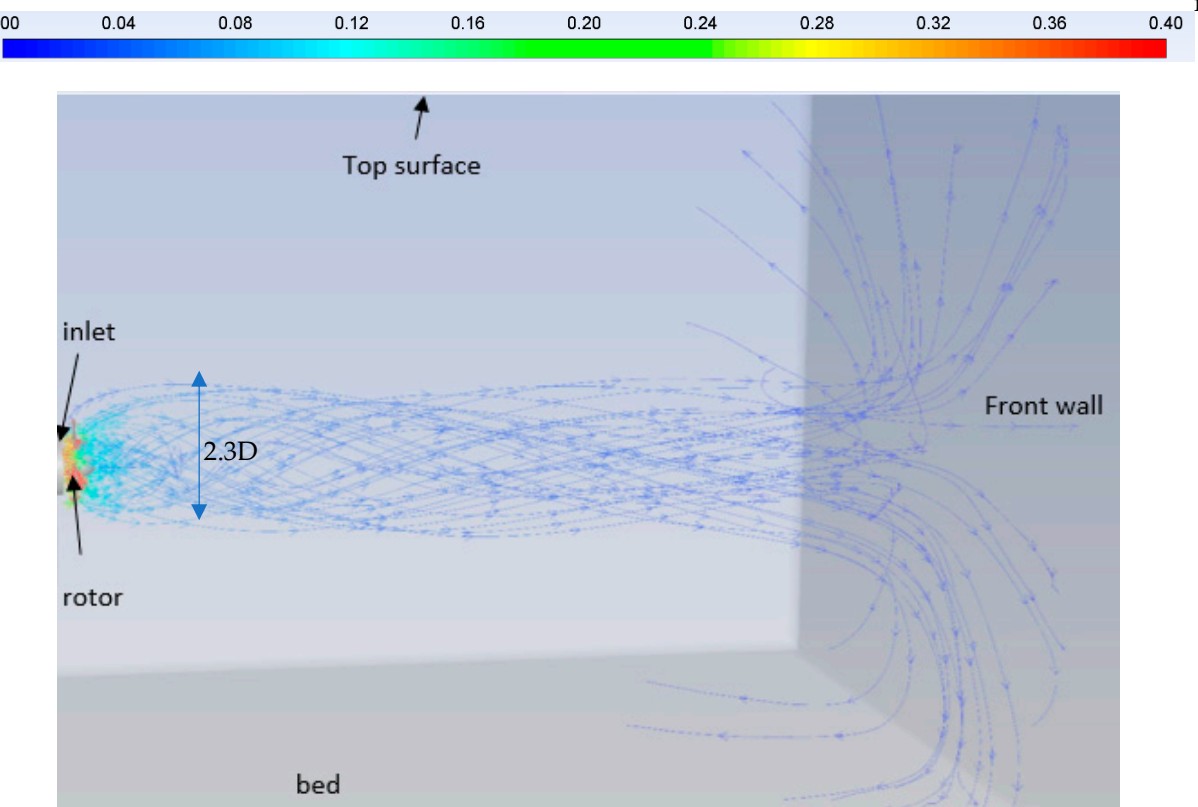

**Figure 14.** The velocity path-lines at $\upsilon_{av}$ = 30 cm/s and $\omega$ = 293.5 rpm.

*5.4. Effect of the Passive Rotor on the Turbulence Intensity*

The turbulent intensity (TI) presents the unresolved unsteadiness and the level of turbulence in a flow over time. It is defined as the ratio of the average fluctuating velocity components ($u_t'$) to the time-averaged absolute "resultant" velocity ($U$) at a certain point in the flow field [44]. Mathematically, the turbulence intensity ($TI$) can be given as:

$$TI = \frac{u_t'}{U}. \tag{4}$$

where:

$$u_t' = \sqrt{\frac{1}{3}\left(u'^2 + v'^2 + w'^2\right)}. \tag{5}$$

$$U = \sqrt{\left(\overline{u}^2 + \overline{v}^2 + \overline{w}^2\right)} \tag{6}$$

The *TI* is a vital quantity for many physical phenomena, including the development of the turbulent boundary layer, heat transfer, and mixing.

Figure 15a illustrates the spatial variations in the turbulence intensity values throughout a vertical plane that passes by the centerline of the pipe outlet and the passive rotor (at $v_{av}$ =30 cm/s and $\omega$ = 293.5 rpm). The figure shows that the turbulence intensity at the pipe outlet varies between 1.19 and 1.95%, with an average value of 1.3%. Slightly further downstream, the passive rotor causes the turbulence intensity to sharply increase to reach significantly higher values than the values downstream of the pipe outlet. It is also noticed that the turbulence intensity substantially decreases as the flow goes far from the passive rotor.

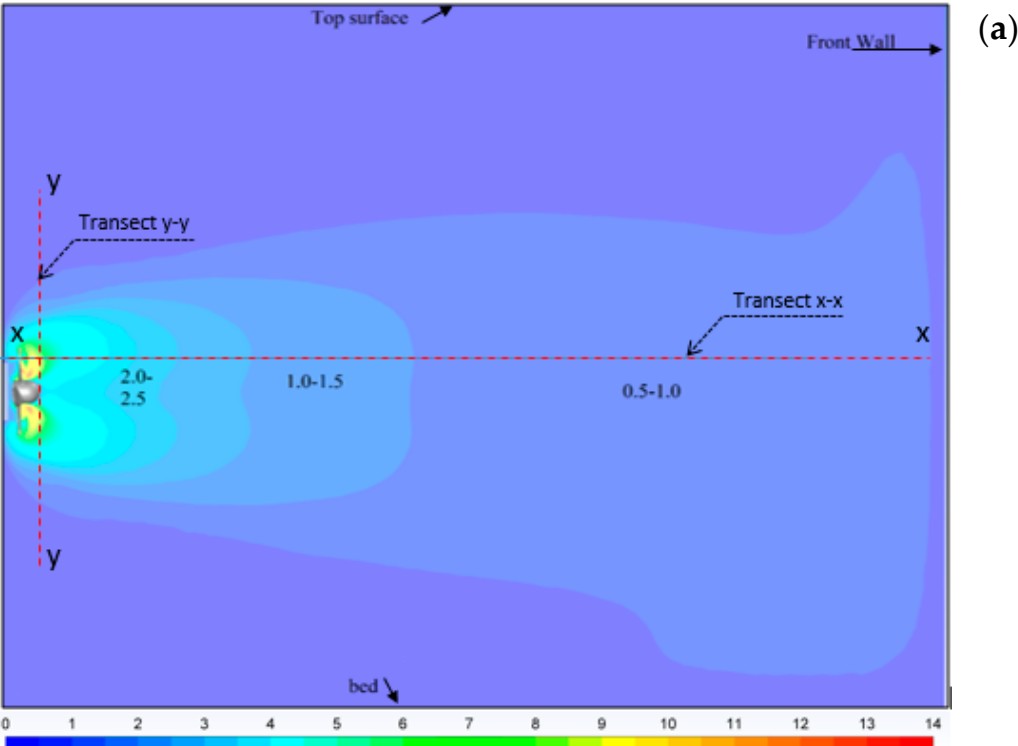

**Figure 15.** *Cont.*

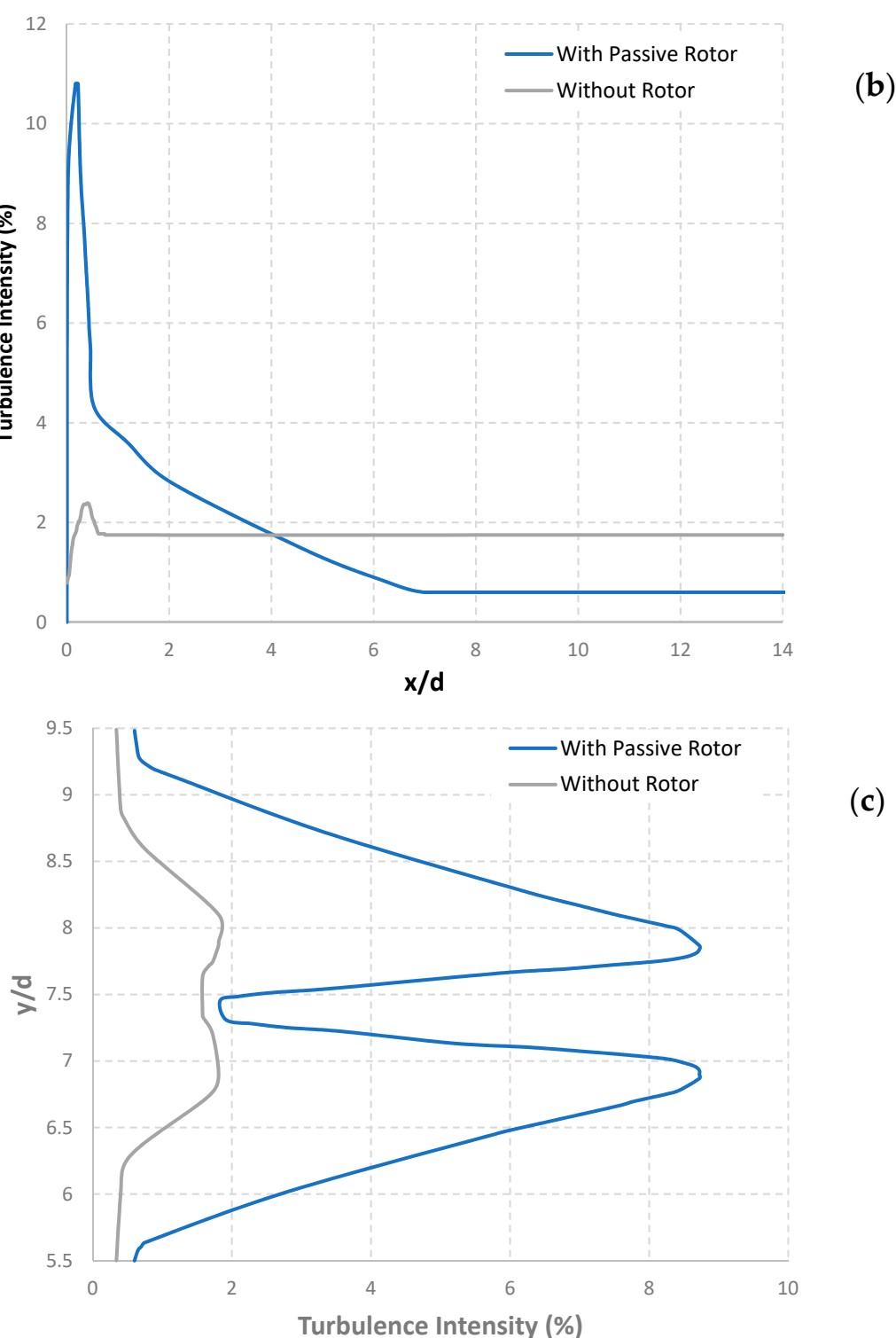

**Figure 15.** Effect of the passive rotor on the turbulence intensity at $v_{av}$ =30 cm/s and $\omega$ = 293.5 rpm. (**a**) Color contour map (vertical plan) of the turbulence intensity downstream of the rotor. (**b**) Spatial variations in turbulent intensity along the x-x transect. (**c**) Spatial variation of TI along the y-y transect.

Figure 15b,c compare the turbulence intensities for the cases of pipe outlets with and without a passive rotor. Figure 15b shows the longitudinal spatial variability throughout the horizontal transect x-x. The vertical position of the transect x-x is selected to pass by

the flow zone, where a high turbulence intensity took place. Figure 15c presents the spatial variability in the vertical direction across transect y-y.

It is noticed that the passive rotor resulted in an increase in the turbulence intensity for the near field zone up to a distance of x ≤ 4.0 d, and the maximum turbulence intensity increased (for the case of "with rotor") to more than five times the corresponding value for the case of "no rotor". However, in the relatively far field zone (x > 4.0 d), the turbulence intensity decreased by more than 50 to 60% compared to the "no rotor" case.

The above-mentioned findings mean that the passive rotor has a significant effect on the turbulence in the near field zone of the pipe outlet; however, this effect vanishes after a distance from the pipe outlet equals 4.0 times the diameter of the pipe outlet.

Using the passive rotor creates a spiral motion, which prolongs downstream of the rotor to a long distance. Tt increases the turbulence intensity considerably. Both the increment of the turbulence intensity and the existence of the spiral motion will help mix the outflow with the ambient water.

Therefore, one can conclude that the passive rotors used in pipe outfalls of desalination plants will produce spiral rotating wakes that will significantly increase the intensity of turbulence downstream of the rotor, resulting in improved mixing in the near field of the rotor, which could be beneficial for thermal dissipation and brine dilution applications. It is also important to emphasize that this improvement applies only to the near field area and is not applicable to the far field.

## 6. Conclusions and Challenges

Passive rotors have been recently introduced and usefully used in a number of water applications. This study proposes a computational modeling workflow based on the ANSYS FLUENT package to simulate the fluid–structure interaction between the emerged water jet from a pipe outlet and a nearby downstream passive rotor. The numerical challenge is to reasonably simulate the temporal variations in the angular speed ($\omega$) of a passive rotor initially at rest and then subjected to time-varying water jet velocity ($\upsilon$). For the numerical model calibration, a lab experiment for a time-varied water flux that emerged from a pipe outlet of a falling head water tank was set, and a passive rotor was added to the pipe outlet. The temporal variation in the angular speed of the passive rotor was measured using the video analysis tracking method, with the help of the Tracker freeware package.

Two computational techniques were investigated; the first is the six-degrees-of-freedom (6DOF), and the second is the sliding mesh (SM). The 6DOF method was applied first since the rotating speed of the computational grid frame is unknown a priori. The objectives of the 6DOF model were: to define the maximum rotor's angular speed, investigate the variations of $\omega$ with time, and deduce a mathematical relation of $\omega$ as a function of the water jet velocity ($\upsilon$). The study has shown that the 6DOF technique is considerably accurate in determining both the maximum and temporal angular speeds, with discrepancies within 3% of the measured values. The 6DOF results indicated a linear relation of the rotor's angular speed ($\omega$) as a function of the average pipe outlet water velocity ($\upsilon$). The SM technique was used as a second step while adopting the obtained w-$\upsilon$ relation from the 6DOF analysis. It has been found that the SM technique results in temporal variations in the rotor's angular speed compared with the physical model measurements. Additionally, the time consumed using the SM technique is considerably less than that required by the 6DOF. Nevertheless, the main limitation of the SM technique is that it requires the modeler to have a priori knowledge of the relationship between the rotor angular speed and the jet velocity. Such limitation does not exist in the case of the 6DOF technique.

The developed computational workflow was used to investigate the following: (1) the effect of the pipe-outlet-to-the-passive-rotor-gap distance on the angular speed of the passive rotor, and (2) the effect of using the passive rotor on the near flow field zone and the water turbulence. The results indicated that the rotor's maximum angular speed decreases as the gap distance between the pipe outlet and the passive rotor increases, following a decline power-law trend. The numerical simulation also presented that the passive

rotor creates a spiral motion of the outlet flow that extends to a relatively long distance (10 times the diameter of the pipe outlet). That spiral motion may help to better mix the incoming water flux inside the water mass. It has also been noticed that using the passive rotor significantly affects the turbulence intensity in the near field zone of the pipe outlet; however, this effect vanishes after the distance from the pipe outlet equals four times the diameter of the pipe outlet.

Using the proposed passive rotors downstream of pipe outlets and outfalls might be challenging in practice. One of the common expected challenges is the sediment and sludges that might be transported with the effluent water, which could break or even remove the rotor from its place. Another challenge could be the expected harm to the fishes swimming nearby the rotor that might take place.

**Author Contributions:** Conceptualization, M.E. and M.F.; methodology, M.F., K.K. and M.E.; software, M.F.; validation, M.E., K.K. and M.F.; formal analysis, M.F., K.K. and M.E.; investigation, M.F., M.E. and K.K.; resources, M.F., M.E. and K.K.; data curation, M.E., K.K.; writing—original draft preparation, M.F. and M.E.; writing—review and editing, K.K. and M.E.; visualization, M.F., M.E. and K.K.; supervision, M.F.; project administration, M.F.; funding acquisition, M.F. All authors have read and agreed to the published version of the manuscript.

**Funding:** This research was supported by the Deanship of Scientific Research, Imam Mohammad Ibn Saud Islamic University, IMSIU, Riyadh, Saudi Arabia, Grant No. 20-13-14-003.

**Institutional Review Board Statement:** Not applicable.

**Informed Consent Statement:** Not applicable.

**Data Availability Statement:** Data are available on request from the authors.

**Acknowledgments:** The authors would like to thank the Deanship of Scientific Research, Imam Mohammad Ibn Saud Islamic University, Saudi Arabia, for supporting and funding this research.

**Conflicts of Interest:** The authors declare no conflict of interest.

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
