# Peer review of "CFD Simulation of a Submersible Passive Rotor at a Pipe Outlet under Time-Varying Water Jet Flux"

_water, doi:10.3390/w14182822_

Round 1
Reviewer 1 Report
This paper reports the investigation on mixing characteristics of the pipe outflow and with the flow caused by the downstream rotor. The main remarks are as follows:
1. In chapter 2.2, the sampling rate of camera is up to 480 fps. Please specify the sampling frequency used in the study. Please provide more detailed information on the set-ups of DAQ system.
2. In chapter 2.3, Figure 4, what is the acceptable value of y+, is it an equal value for all the near wall regions, especially those on the stationary walls?
3. Please provide a figure, which show the mesh in the focused region.
4. In Figure 6, how to make sure that the number of grids is enough for the calculation? An additional question is that whether the deformation of rotor blade is taken into account?
5. From Line 378 to Line 383, the criteria of selecting the turbulence model, the velocity in the domains of simulation should be also taken into consideration.
6. In figure 9, the error of measurements should also be added. It is also recommended to add error bars to the related figures.
7. In figure 13, it is recommended to specify the reason of selecting the vertical position of x-axis. The curves are not smooth, which may be attribute to the insufficient points on the sample-lines.
8. The results are not enough. It is recommend to analyze the flow carefully and reveal the physics.
9. The conclusion and challenge should be re-organized. The physics of flow should be carefully analyzed.
10. There are some expression errors, such as [20] used a glyph script, need to be modified. The quality of the figures should be improved because they are not clear.
Author Response
please use the pdf version of the reply to the first reviewer

Reviewer 2 Report
This paper presents an interesting study on the fluid-structure interaction between the water and the rotor by using a FLUENT based computational workflow, so as to estimate the temporal variation of the angular speed (ω) of a passive rotor initially at rest and then subjected to time-varying water velocity (Ê‹). Overall this research topic and content fall within the scope of this special issue and journal. However, the following issues should be FULLY addressed by the authors before the consideration of publication:
1. For readers to quickly catch your contribution, it would be better to highlight major difficulties and challenges, and your original achievements to overcome them, in a clearer way in abstract and introduction.
2. The manuscript could be substantially improved by relying and citing more on recent literature about real case studies of applications of CFD models or related topics, which has been widely studied by peers in recent years and thus can be easily found from many journal websites, such as the following:
- (2021) The influence of the blade tip shape on brownout by an approach based on computational fluid dynamics, Engineering Applications of Computational Fluid Mechanics, 15:1, 692-711, DOI: 10.1080/19942060.2021.1917454.
- (2020) Effects of diameter and suction pipe opening position on excavation and suction rescue vehicle for gas-liquid two-phase position, Eng. Appl. of Comput. Fluid Mech., 14:1, 1128-1155, DOI: 10.1080/19942060.2020.1813204.
- (2020) Dynamic flow behavior and performance of a reactor coolant pump with distorted inflow, DOI: 10.1080/19942060.2020.1748720.
3. In the results section, an in-depth discussion should be made for enhancing the practical implication of the results and findings from current study, which will be more useful than that for only performing the statistical data collection and analysis.
4. In the conclusion section, the limitations of this study, suggested improvements and future direction of this work should be highlighted.
Author Response
Reviewer General Comment: This paper presents an interesting study on the fluid-structure interaction between the water and the rotor by using a FLUENT based computational workflow, so as to estimate the temporal variation of the angular speed (ω) of a passive rotor initially at rest and then subjected to time-varying water velocity (Ê‹). Overall, this research topic and content fall within the scope of this special issue and journal. However, the following issues should be FULLY addressed by the authors before the consideration of publication:
Reply: The authors would like to thank the reviewer for his comments, and they are sure that their comments will result in a better-quality manuscript. The paragraphs to follow address point by point the author’s reply to these comments.
Reviewer Comment 1: For readers to quickly catch your contribution, it would be better to highlight major difficulties and challenges, and your original achievements to overcome them, in a clearer way in abstract and introduction.
Reply: Noted, and the introduction is revised accordingly.
Reviewer Comment 2: The manuscript could be substantially improved by relying and citing more on recent literature about real case studies of applications of CFD models or related topics, which has been widely studied by peers in recent years and thus can be easily found from many journal websites, such as the following:
Reply: done as requested
Reviewer Comment 3: In the results section, an in-depth discussion should be made for enhancing the practical implication of the results and findings from current study, which will be more useful than that for only performing the statistical data collection and analysis.
Reply: The analysis has been extended, and some numerical runs have been conducted to study the effect of the gap distance between the rotor and the pipe outlet on the rotor angular speed.
Also, the practical implication of the results was discussed.
Reviewer Comment 4: In the conclusion section, the limitations of this study, suggested improvements and future direction of this work should be highlighted.
Reply: Some challenges have been discussed in the conclusion section. The limitation of the study and
suggested improvements and future direction are added to the conclusion as requested.

Round 2
Reviewer 1 Report
This paper reports the investigation on mixing characteristics of the pipe outflow and with the flow caused by the downstream rotor. The main remarks are as follows:
1. In chapter 3.1, Figure 5, what is the acceptable value of y+, is it an equal value for all the near wall regions, especially those on the stationary walls?
2. Please provide a figure, which show the mesh in the focused region.
3. In Figure 7, how to make sure that the number of grids is enough for the calculation?
4. From Line 451 to Line 456, the criteria of selecting the turbulence model, the velocity in the domains of simulation should be also taken into consideration.
5. In figure 10, the error of measurements should also be added. It is also recommended to add error bars to the related figures.
6. In figure 15, it is recommended to specify the reason of selecting the vertical position of x-axis.
Author Response
The authors would like to thank the reviewers for their comments and they are sure that their comments will result in a better-quality manuscript. The paragraphs to follow address point by point the author’s reply to these comments. Kindly be noted also that the corresponding changes in the manuscript are highlighted in red. Note that some of the comments might overlap with the other reviewers’ comments.
- In chapter 3.1, Figure 5, what is the acceptable value of y+, is it an equal value for all the near wall regions, especially those on the stationary walls?
Reply: noted and revised (The maximum Y+ is less than 5 for the rotor with an average value equal to 2.34, while the maximum Y+ is less than 1 for all other fixed walls with an average value equal to 0.6)
- Please provide a figure, which show the mesh in the focused region.
Reply: noted and revised. The below figure has been added as figure 7.c
3. In Figure 7, how to make sure that the number of grids is enough for the calculation?
Reply: the authors used the Grid Convergence Index (GCI), which is a relatively new discretization error estimation technique. The GCI is calculated to answer whether the adopted mesh in the simulation is refined enough or not. Please refer to section 4.2 in the paper and to the below reference
Celik, I.B., Ghia, U., Roache, P.J. and Freitas, C.J., 2008. Procedure for estimation and reporting of uncertainty due to discretization in CFD applications. Journal of fluids Engineering-Transactions of the ASME, 130(7).
4. From Line 451 to Line 456, the criteria of selecting the turbulence model, the velocity in the domains of simulation should be also taken into consideration.
Reply: noted and revised. The criteria of selecting the turbulence model, the maximum velocity (0.6 m/s) has been added
5. In figure 10, the error of measurements should also be added. It is also recommended to add error bars to the related figures.
Reply: Figure 10a is revised as requested (as shown below) to include the error bar related to the measurement. Please note that the estimated error in the measurement is of the order of 1%. Figure 10a is revised accordingly in the text
6. In figure 15, it is recommended to specify the reason of selecting the vertical position of x-axis.
The vertical position of the transect x-x is intentionally selected to pass by the flow zone where high turbulence intensity took place. As requested, the above-mentioned justification is added to the text for clarity.
Reviewer 2 Report
The paper has been greatly improved, which is now acceptable for publishing on this journal.
Author Response
Many thanks for your valuable comments